# The Potential of the Flavonoid Content of *Ipomoea batatas* L. as an Alternative Analog GLP-1 for Diabetes Type 2 Treatment—Systematic Review

**DOI:** 10.3390/metabo14010029

**Published:** 2023-12-31

**Authors:** Ni Kadek Santi Maha Dewi, Yan Ramona, Made Ratna Saraswati, Desak Made Wihandani, I Made Agus Gelgel Wirasuta

**Affiliations:** 1Doctoral Study Program, Faculty of Medicine, Udayana University, Denpasar 80232, Indonesia; santimahadewi@unud.ac.id; 2Pharmacy Department, Faculty of Mathematic and Natural Science, Udayana University, Kampus Bukit Jimbaran, Denpasar 80361, Indonesia; 3Biology Department, Faculty of Mathematic and Natural Science, Udayana University, Kampus Bukit Jimbaran, Denpasar 80361, Indonesia; yan_ramona@unud.ac.id; 4Department of Internal Medicine, Faculty of Medicine, Udayana University, Denpasar 80232, Indonesia; ratnasaraswati@unud.ac.id; 5Department of Biochemistry, Faculty of Medicine, Udayana University, Denpasar 80232, Indonesia; dmwihandani@unud.ac.id; 6Forensic Sciences Laboratory, Institute of Forensic Sciences and Criminology, Udayana University, Kampus Bukit Jimbaran, Denpasar 80361, Indonesia

**Keywords:** anti-diabetic, GLP-1, *Ipomoea batatas* L. flavonoid, pharmacology

## Abstract

*Ipomoea batatas* L. (IBL) has gained significant popularity as a complementary therapy or herbal medicine in the treatment of anti-diabetes. This review seeks to explore the mechanism by which flavonoid compounds derived from IBL exert their anti-diabetic effects through the activation of GLP-1. The review article refers to the PRISMA guidelines. In order to carry out the literature search, electronic databases such as Science Direct, Crossref, Scopus, and Pubmed were utilized. The search query was based on specific keywords, including Ipomoea batatas OR sweet potato AND anti-diabetic OR hypoglycemic. After searching the databases, we found 1055 articles, but only 32 met the criteria for further review. IBL contains various compounds, including phenolic acid, flavonols, flavanols, flavones, and anthocyanins, which exhibit activity against anti-diabetes. Flavonols, flavanols, and flavones belong to a group of flavonoids that possess the ability to form complexes with AlCl_3_ and Ca^2+^. The intracellular L cells effectively retain Ca^2+^, leading to the subsequent release of GLP-1. Flavonols, flavones, and flavone groups have been found to strongly interact with DPP-IV, which inhibits the degradation of GLP-1. The anti-diabetic activity of IBL is attributed to the mechanism that effectively increases the duration of GLP-1 in the systemic system, thereby prolonging its half-life.

## 1. Introduction

The rise in diabetes cases worldwide can be attributed to several factors, including aging populations, population growth, and the increasing prevalence of the disease among different age groups [1]. According to projections made by the World Health Organization, DM is anticipated to rank as the seventh highest cause of mortality on a global scale by the year 2030. This forecast is based on the significant increase in the prevalence of this disease in recent times. In 2015, the International Diabetes Federation reported a staggering 415 million cases of diabetes worldwide. Alarming as it may be, this number is projected to climb even higher, reaching an estimated 642 million cases by the year 2040, according to their latest forecasts [2].

A total of 1055 publications have been identified to examine/report the potential of *Ipomoea batatas* L. (IBL) as an alternative method for diabetes treatment (Figure 1). These publications were categorized into 12 distinct research clusters, which were distinguished by the size of their nodes. The majority of the extensive cluster focused on *Ipomoea batatas* L. as an anti-diabetic and antioxidant, while a smaller portion delved into only one group, reporting on the mechanism of Glucagon-like peptide-1 (GLP-1) on IBL.

All the published articles searched for our review mentioned that IBL contains various active compounds, including flavonoids, anthocyanins, phenolic acids, caffeoyl derivatives, triterpenoids, and alkaloids. Each compound is still discussable regarding its pharmacological mechanism. Recent reports indicated that flavonoid compounds have been demonstrated to have efficacy against diabetes mellitus (DM). The occurrence effect of flavonoid consumption to reduce DM is signaled by several signaling pathways, namely glucose transporters, liver enzymatic, tyrosine kinase inhibition, AMPK, PPARγ, and NF-κB [3,4,5]. At present, researchers have identified 27 IBL cultivars that possess anti-diabetic properties. These compounds exhibit pharmacological effects and target multiple sites of action, namely the pancreas, liver, skeletal muscle, and adipose tissue [6]. IBL’s therapeutic effect on diabetes is achieved through a multifaceted pharmacological mechanism that acts on various chemical and pharmacological sites.

The compounds that have been identified were predicted to work in the gastrointestinal (GI) tract by α-glucosidase inhibition through the degradation of polysaccharides into monosaccharides and the secretion of GLP-1. It can lead to a reduction in gastric emptying, a decrease in gastrointestinal motility, and an increase in insulin secretion [7,8]. These findings also illustrated the effect on lowering blood pressure and cholesterol, both of which were identified as risk factors for DM based on the resulting outcome. A study IBL extract could induce the release of GLP-1, thereby making it an effective anti-diabetic agent [9]. This review article explores the potential and action mechanism of flavonoid compounds in IBL that can trigger GLP-1 activation.

## 2. Materials and Methods

### 2.1. Search Strategy

Consistent with the PRISMA criteria, we implemented a search strategy to conduct our study [10]. Searching for relevant articles, we conducted a systematic review of IBL’s therapeutic potential in anti-diabetic treatment. We used several selected databases, such as Science Direct, Scopus, Crossref, and Pubmed, to search extensively for the literature. The keywords encompassed in the search query were (1) Ipomoea batatas OR sweet potato AND (2) anti-diabetic OR hypoglycemic.

### 2.2. Inclusion Criteria

We selected research articles that focused on the effects, anti-diabetic potentials, phytochemical compounds, and signaling mechanisms of *Ipomoea batatas***.** English-written articles that relied on in vitro and in vivo studies formed the core of these research papers. The selected studies evaluated at least three essential measures: (1) *Ipomoea batatas*, (2) phytochemical compounds, and (3) the signaling mechanisms involved.

### 2.3. Exclusion Criteria

In order to proceed with our review, we excluded conference papers, thesis dissertations, review articles, papers published for conferences, and manuscripts without abstracts or that did not meet the inclusion requirements. Studies that looked into the relationship between *Ipomoea batatas* and other diseases were not part of this analysis.

### 2.4. Data Extraction and Management

To compile the articles for this study, a reference manager called Zotero was utilized. By examining the publications that met the inclusion criteria, we proceeded with our analysis. The information gathered included the (1) type/cultivar, (2) part of the plant, (3) identified compound, (4) bioactive compound, (5) site of action, and (6) anti-diabetic mechanism pharmacology activity of IBL.

### 2.5. Data Extraction Strategy

In this literature review, the outcomes of various in vitro and in vivo studies examining the influence of IBL on type 2 DM were presented. Section 3 and Section 4 describe the results of the reports regarding the phytochemicals involved. In Section 5, we analyze the findings, considering the sites of action and pharmacological mechanisms of diabetes treatment.

## 3. Results

### The Literature Search

The literature search identified 1055 articles relevant to the topic (Figure 1). Duplications were detected and removed, totaling 44 articles. Based on screening the titles and abstracts, 865 articles were removed. Then, 125 articles were further excluded based on the results of the screening of the inclusion criteria mentioned above. A total of 31 appropriate articles were reviewed in more depth in Table 1. The successful data extraction is shown in the flow diagram displayed in Figure 2.

## 4. Discussion

### 4.1. Type or Cultivar

IBL has several varieties. These varieties are differentiated based on tuber color, skin color, leaf color, texture, and size. The number of cultivars identified in this journal was 27, with several variations, such as orange [16,22,28,33], purple [12,14,19,21,24,25,27,33,34,40], and white IBL [8,9,14,15,16,18,21,24,27,28,29,30,34,36,37]. Differences in the cultivars will affect the phytochemical contents and their anti-diabetic activity.

### 4.2. Parts of Plant and Phytochemical Identified of IBL

Many reports mentioned that parts of IBL used for its anti-diabetic effects include the leaves, tubers, and tuber skin. Each part has different chemical compositions. In purple IBL tubers, the anthocyanin content commonly used as a marker is higher than in its leaves. The concentration of anthocyanins is also greater in purple IBL compared to white or orange IBL. The phenolic acid content, such as 3,4,5-Tricaffeoylquinic acid, chlorogenic acid, caffeic acid, Isochlorogenic acid C, Isochlorogenic acid A, and caffeoyl acid derivative [10,11,15,20], flavonoid groups such as C3R, C3G, C35G, cyanidin 3-caffeoyl-p-hydroxybenzolsophoroside-5-glucoside, P3G, peonidin 3-caffeoyl-p hydroxybenzoyl sophoroside-5-glucoside, peonidin 3-O-[2-O-(6-O-E-feruloyl-β-D-glucopyranosyl)-6-O-E-caffeoyl-β-D-glucopyranoside]-5-O-β-D-glucopyranoside [21,27], quercetin, epicatechin, protocatechualdehyde, rutin, kaemferol, isoquercitrin, and jaceosidin, have been identified in such type of potato. They directly serve as effective anti-diabetic agents [10,11,22,35]. Flavonol, a subclass of flavonoids, is extensively found in various natural sources. Flavonols, such as quercetin and epicacthecin, demonstrated their potential for increasing GLP-1 secretion in a tissue culture of GLUTag cells [41]. Ground triterpenoid, such as trans-N-feruloyltyramine, *trans*-*N*-(*p*-coumaroyl) tyramine, *cis*-*N*-feruloyltyramine, and 7-hydroxy-5-methoxycoumarin, and alkaloid groups such as Indole-3-carboxaldehyde also have potential as anti-diabetic agents [16].

#### 4.2.1. Site of Action

GLP-1 is produced in the intestine through the posttranslational processing of proglucagon. The L cells, primarily found in the colon and ileum, are types of open-type epithelial cells that directly interact with nutrients in the intestinal lumen. The level of GLP-1 in circulation quickly rises due to nutrients such as carbohydrates, fats, proteins, and dietary fiber [42]. Glucose is taken up through GLUT-2, fructose through GLUT-5, and SCFAs are absorbed and metabolized intracellularly. The GLP-1 secretion is induced by the closure of KATP channels, which is a result of carbohydrate uptake through SGLT1, GLUT-2, and GLUT-5. Through intracellular metabolism, cell membrane depolarization is triggered, leading to the production of ATP and the closure of KATP channels. Additionally, this process facilitates the opening of voltage-gated Ca^2+^ channels. Furthermore, the uptake of free amino acids and peptides also induces depolarization and the subsequent activation of voltage-gated Ca^2+^ channels (VGCCs). The activation of VGCCs is brought about by the coupled transport of Na+ for amino acids and PepT1 for peptides, leading to the stimulation of GLP-1 secretion. The release of GLP-1 is triggered by the influx of extracellular Ca^2+^ and the release of Ca^2+^ from intracellular reservoirs, resulting in additional depolarization and the subsequent activation of the exocytotic machinery [43,44]. The excessive production of GLP-1 within the cells results in its dispersion across the entire systemic system. It then attaches to the GLP-1R receptor found in different organ tissues, including the skeletal muscle, adipose tissue, liver, pancreas, and gastrointestinal tract. The binding of GLP-1 to the liver results in a decline in glucose production, while concurrently promoting an elevation in glucose uptake within adipose tissue and muscle [45,46].

Various flavonoid compounds, including hispidulin, epicatechin, quercetin, C3G, 5,7-dihydroxy-6-4-dimethoxyfavanone, and homoesperetin-7-rutinoside, have demonstrated their ability to enhance GLP-1 release both in vitro and in vivo. GLP-1 stimulation has been demonstrated in GLUTag cells when exposed to epicatechin, C3G, and hispidulin. Homoesperetin-7-rutinoside has also exhibited stimulation through molecular docking [47]. The chelation of Ca^2+^ by quercetin, similar to the chelation of AlCl_3_, has been documented in various studies. Complexes between quercetin and metals are formed at the ortho positions O3/O4, O4/O5, and O3′/O4′, as detailed by numerous reports in the literature (Figure 3) [48]. Compounds belonging to the flavonol, flavanol, and flavones groups exhibit activity against AlCl_3_, resulting in a yellow color change. It is anticipated that these compounds will operate via a similar mechanism against Ca^2+^. These compounds have the ability to prolong the half-life of GLP-1, leading to an elevation in insulin release and a modification in glucose absorption from systemic to cellular in the form of glycogen [49]. Furthermore, the A or B rings of flavonoids are capable of interacting with AlCl_3_ through their ortho-dihydroxyl groups, resulting in the formation of complexes that are susceptible to acid [50].

Increased oxidative stress is largely responsible for the development and improvement of DM and its complications. Pancreatic islets have low expression levels of antioxidant enzymes, which make them more vulnerable to oxidative damage. Biomarkers such as MDA are used to assess oxidative stress, with increased levels indicating higher levels of lipid peroxidation. Oxidative stress in diabetes is reduced by GLP-1, which activates the cAMP, PI3K, and PKC pathways through receptors and Nrf-2. This also increases the antioxidant capacity. Conversely, oxidative stress can be reduced by suppressing ROS through radical scavenging and chelating mechanisms, thus protecting pancreatic beta cells [51].

#### 4.2.2. Gastrointestinal Tract

##### Regulation of Carbohydrate Metabolism

The inhibition of α-glucosidase influenced the capability of the small intestine to inhibit the absorption of carbohydrates. This particular enzyme inhibited the conversion of complex carbohydrates and was unable to be assimilated into simple carbohydrates [52]. The IC_50_ values of all the compounds were investigated to be in the range of 4.46 µM to 64.14 µM, which were observed in various studies on the efficacy of α-glucosidase inhibitors in comparison to acarbose. Ethyl caffeate could inhibit α-glucosidase more effectively (about 6.77 times more than acarbose). These flavonoid compounds had a potent inhibitory effect on α-glucosidase, such as rutine, isoquercitrin, quercetin, kaempferol, and hyperoside [12]. Compared to acarbose, Trans-N-(p-coumaroyl)tyramine, 3,4,5-Tricaffeoylquinic acid, trans-N-feruloyltyramine, and cis-N-feruloyltyramine inhibited α-glucosidase 37.9, 36.6, 18.7, and 11.8 times more effectively, respectively. Quercetin-3-O-glucosidase and 7-Hydroxy-5-methoxycoumarin, with respective IC_50_ values of 22.38 ± 1.73 µM and 64.14 ± 9.23 µM, also had good activity [16]. Chlorogenic acid was also predicted to have inhibition activities on α-glucosidase and tyrosinase. The docking results also supported that Chlorogenic acid binds to the active site of α-glucosidase and can act in a reversible competitive manner through hydrogen bonds [19]. Peonidin 3-O-[2-O-(6-O-E-feruloyl-β-D-glucopyranosyl)-6-O-E-caffeoyl-β-D-glucopyranoside]-5-O-β-D glucopyranoside showed that a potent maltase inhibition was preferred over sucrase inhibition [27]. The aqueous fraction contained an acidic glycoprotein (IC_50_ 53 µg/mL) that exhibited anti-diabetic properties and was predicted to have a mechanism of α-glicosidase inhibition, which is a significant breakthrough for the care of DM. This finding is particularly noteworthy when compared to acarbose [14].

Ethyl caffeic had an α-amylase activity that was 13.1 times stronger than acarbose. This was possibly due to the contribution of the OH group of the ethyl caffeic binding to the enzyme. The mixed-type inhibition of α-amylase by chlorogenic acid occurred as a result of the binding of chlorogenic acid to amino acid residues near the active site through hydrogen bonds. This binding process led to a modification in the secondary structure of the enzyme’s protein, thereby inhibiting its activity [12]. C3G, C3R, C35G, and P3G had IC_50_ values of 0.024 ± 0.003 (mM), 0.040 ± 0.007 (mM), 0.031 ± 0.007 (mM), and 0.075 ± 0.007 (mM), respectively, for porcine pancreatic α-amylase inhibition These four anthocyanin compounds were proven in silico, and the active side of the compound was thought to be mediated by interacting with the carboxylate group of GLU233. C3G was the most potent inhibitor of the four compounds, with a low Ki of 0.0014 mM. The Ki values of the other compounds were much higher: 0.019 nM for C3R, 0.020 nM for C35G, and 0.045 nM for P3G. The inhibition activity was observed to be conferred by the shared key side chain GLU233 [21]. Each compound induced absorption that exclusively took place as fiber and was subsequently eliminated through the gastrointestinal tract.

##### Increased Insulin Secretion

GLP-1, a hormone known as incretin, is released by intestinal endocrine L cells, which exhibit an effect of eaten food. The enhancement of GLP secretion in the gastrointestinal tract can lead to an improvement in insulin secretion, a decrease in GI motility, and a delay in gastric emptying. Various studies have identified compounds belonging to the phenolic acid and flavonoid groups that exhibit potential for augmenting GLP-1 secretion [7,8].

The most potent chlogenic acid derivative was 3,4,5-Tricaffeoylquinic acid, which increased GLP-1 secretion around 10-fold compared to the sulfonylure control tested on GLUTag cells. The same effect was also produced during the in vivo test, where the treatment group produced more GLP-1 secretion compared to the sulfonylurea control group. Hence, GLP-1 has the ability to maintain glycemic balance without the potential of inducing hypoglycemia. The elevation of cAMP concentrations may lead to the stimulation of GLP-1 secretion in both in the vivo model and a cell line. The noticed elevations in GLP-1 production in L cells were caused by the activation of PKA and cAMP in the in vitro model using GLUTag cells [9]. Cyanidin 3-O-glucoside (C3OG) and epicatechin functioned through the activation of the cAMP/PKA and ERK ½ pathways. Ca^2+^ chelation’s mechanism of action led to an elevation in the intracellular Ca^2+^ concentration, consequently resulting in an augmentation of GLP-1 secretion. The activity of this compound seemed to be supported by the appearance of a 3′4′ catechol group in the B ring, which was considered a crucial chemical structural component [41]. Hispidulin was also recognized as an effective anti-DM agent. Notably, the study demonstrated that hispidulin treatment led to an increase in the intracellular cAMP levels in L cells [47].

An alternative approach that has shown clinical benefits involves inhibiting DPP-4 to prolong the duration of GLP-1. Flavonoids, namely flavonol, flavanol, and flavone (Figure 3), displayed varying degrees of inhibitory activity, which were dependent on their concentration. Notably, narcissoside, myricetin, C3OG, hyperoside, and isoliquiritigenin demonstrated higher inhibitory activities. An analysis of the relationship between the structure and activity revealed that incorporating hydroxyl groups at positions C3′, C4′, and C6 in the flavonoid structure enhanced the effectiveness of DPP-4 inhibition. However, the addition of a hydroxyl group at position three of ring C in the flavonoid configuration was discovered to be disadvantageous for suppression. Additionally, the methylation of the hydroxyl groups at positions C3’, C4′, and C7 of the flavonoid conformation tended to reduce the inhibitory activity against the DPP-4 enzyme. Furthermore, it was determined that the presence of a 2,3-double bond and a 4-carbonyl group on ring C of the flavonoid configuration was crucial for achieving the inhibitory effect [53].

#### 4.2.3. Pancreas

##### Inhibiting Apoptosis Beta Cell and Recovering the Islet Structure through Protective Cell Beta

Pancreatic beta cells have a crucial function in regulating glucose balance and serve as the primary producers of insulin. Their role encompasses the synthesis, storage, and secretion of insulin [54]. The research findings indicated that the IC_50_ value for total compound fell within the range of 9.69 ± 0.03 µM to IC_50_ 125 ± 0 µM for Vit C. This particular range demonstrated a significant efficacy for conferring natural antioxidant properties to the compound. An elevation in ROS was the sole factor responsible for harm to the beta cells. Antioxidants in abundance are believed to have the potential to diminish ROS, leading to the recovery of islet beta cells. This recovery process will enhance the insulin secretion process through protective cell beta.

Giving SPLP (sweet potato leave phenol) leaf extract from Beijing to T2DM mice for 4 weeks showed a recovery of pancreatic tissue with an increased area and complete islet structure, increased mass, and clear borders. Giving SPLP treatment inhibited beta cell apoptosis. Phenolic compounds, including 1-caffeoylquinic acid, chlorogenic acid, caffeac acid, and 3,4,5-tricaffeoylquinic acid, are believed to have the competence to govern the process of beta cell regeneration. The IC_50_ value of 3,4,5-Tricaffeoylquinic acid indicated a radical scavenging activity that was 10.8 times more potent than the control substance, cevitamic acid. Subsequently, the other compounds exhibiting greater strength than ascorbic acid included Isochlorogenic acid C, Isochlorogenic acid A, Isochlorogenic acid B, Caffeic acid ethyl ester, and Caffeic acid. Additional research has documented the presence of ABTS-induced antioxidant activity. Ethyl caffeate and 3,4,5-Tricaffeoylquinic acid have been identified as exhibiting commendable activity, with IC_50_ values comparable to that of ascorbic acid. The order of antioxidant potency, in terms of descending activity, can be arranged as follows: Isochlorogenic acid C exhibited the highest potency, followed by Chlorogenic acid, Neochlorogenic acid, caffeic acid, Cryptochlorogenic acid, Isochlorogenic acid A, 1-Caffeoylquinic acid, Isochlorogenic acid B, esculin, and finally 7-hydroxycoumarin.

Protocatechualdehyde exhibited the most potent antioxidant activity, surpassing that of ascorbic acid by a factor of 2.32. Quercetin exhibited the highest antioxidant activity among the flavonoid group, with a remarkable 4.14-fold increase compared to ascorbic acid. Following closely, kaemferol and jaceosidin demonstrated a commendable ABTS radical scavenging activity, which was nearly on par with cevitamic acid. In the DPPH radical scavenging test, protocatechualdehyde displayed the most potent DPPH activity, surpassing ascorbic acid by a significant factor of 2.69. Ethyl caffeate and 3,4,5-tricaffeoylquinic had a good DPPH activity. The descending order of DPPH antioxidant activity, ranging from high to low, is as follows: caffeic acid, isochlorogenic acid C, chlorogenic acid, 1-caffeoylquinic acid, neochlorogenic acid, isochlorogenic acid A, cryptochlorogenic acid, isochlorogenic acid B, 7-hydroxycoumarin, and esculin. Quercetin exhibited a significantly higher DPPH radical scavenging activity that was 1.8 times greater than that of cevitamic acid. Notably, isoquercitrin, hyperoside, kaempferol, and routine also demonstrated commendable scavenging capabilities. The remarkable DPPH radical scavenging capacity of caffeic acid could be attributed to its 1,2-phenolic diol group, as well as its conjugation involving the C=C and C=O bonds. Another mechanism using the FRAP method was that protocatechualdehyde had a 2.22 stronger activity than cevitamic acid. Ethyl caffeate, caffeac acid, 3,4,5-tricaffeoylquinic acid, and Isochlorogenic acid also had a good activity. For the flavonoid compounds, namely quercetin produced a stronger reducing power activity of 1.54 compared to vitamin C. Hyperoside, kaempferol, isoquercitrin, and rutine exhibited commendable FRAP capacities [11,16,55]. The IC_50_ values of SPLP were consistent with a prior study, indicating a more potent oxidative stress protection compared to polyphenols derived from tea and grape seed [56].

The significant increase in the enzymatic antioxidants SOD and GSH-Px were correlated with an antioxidant activity to reduce ROS. A decrease in ROS will cause the recovery of pancreatic beta cells. The compound suspected in this process was protein-bound anthocyanin from tuber. Anthocyanins that were identified from purple IBL and possibly bound to protein were peonidin-3-cyanidin-3-sophoroside-5-glucoside and sophoroside-5-glucoside [15,24,57]. Additional research indicated that Cyanidin 3-caffeoyl-p-hydroxybenzoyl-sophoriside-5-glucoside exhibited the most potent antioxidant properties among the various anthocyanin compounds. This finding was determined through the utilization of ascorbic acid and the ABTS and DPPH methods, as stated in the alternative investigations. Furthermore, peonidin 3-caffeoyl sophoroside-5-glucoside exhibited superior antioxidant activities compared to peonidin3-(6″-caffeoyl-6″-feruoyl sophoroside)-5-glucoside and peonidin3-caffeoyl-p-hydroxybenzoyl-sophoroside-5-glucoside [33].

Tuber extract and orange-fleshed IBL leaves as beta cell protectors can reduce lipid peroxidation through a radical scavenging mechanism and the results of measuring the antioxidant activity with FRAP and TEAC can significantly reduce ROS. The antioxidant capacity values using FRAP were, respectively, 299.8 ± 2.5 and 296.9 ± 7.4 (µM AAE/mg protein) and testing using the TEAC method were 127.9 ± 2.10 and 126.3 ± 2.51 (µM TE/mg Protein). The results of this test were greater than standard ascorbic acid, namely 271.0 ± 4.17 (µM AAE/mg protein) and 107.2 ± 1.68 (µM TE/mg protein). The compounds predicted to be contained in the extract that had radical scanning activity were caffeic acid, hyperoside, protocatechuic acid, quercetin, routine, and vanillic acid [22]. Caffeic acid was reported to be a compound with a potent antioxidant activity [58,59]. MAE (microwave assisted extraction) leaves of purple IBL cultivar antin-3, which were predicted to contain the anthocyanin group, produced an IC_50_ value of 61.91 ± 1.11 ppm [37].

This compound worked directly in increasing insulin secretion through the repair of islet beta cells. This compound was responsible for scavenging reactive oxygen species (ROS) and elevating the AMP/ATP ratio within beta cells. The alteration in the AMP/ATP ratio triggered the activation of mitochondrial targets, leading to the induction of mitogenesis and the stimulation of insulin secretion [60].

##### Suppression of the Anti-Inflammatory Pathway

The compounds identified from Bandungan, Java, Indonesia, were anthocyanin, catechin, quercetin, proanthocyanin, and caffeic acid. These compounds were predicted to work in the pancreas by inhibiting the anti-inflammatory mechanisms [20]. The administration of the leaf extract at a dose of 2.5 g/kgBW for a duration of 14 days demonstrated a noteworthy 50% augmentation in pancreatic islet cells in comparison to the control group. Conversely, the administration of caiapo at a dose of 5 g/KgBW over a period of 8 weeks substantially enhanced the beta cell mass by a two-fold increase when compared to the untreated diabetic control subjects (*p* < 0.05). The results indicated that increased dosages of the extract could potentially result in a more significant restoration of islet beta cells. It is important to highlight that the extract was thought to comprise quercetin, chlorogenic acid, caffeic acids, and their derivatives. The anti-inflammatory characteristics of these compounds are believed to play a crucial role in diminishing inflammation by inhibiting inflammatory mediators. Quercetin is known to inhibit tyrosine kinase activity, which has been shown to be anti-diabetic. The regulation of quercetin effects through the inhibition of the NF-κB activation of beta cells also helps to improve glucose-stimulated insulin secretion [14,61,62]. Apart from that, the content of chlorogenic acid, caffeic acids, and their derivatives is thought to be able to inhibit the JNK, P38 MAP, and NF-κB pathways and is also associated with various inflammatory mediators, including IL-6, CRP, and TNF-α. It has also been reported that inhibiting oxidative stress may also induce hypoglycemic effects [35].

#### 4.2.4. Liver

##### Improving Insulin Secretion and Insulin Sensitivity by Reducing Glucose Synthesis

The primary structures of other acylated anthocyanins, namely C3S5G and P3S5G, are anticipated to play a vital role in improving glucose absorption and increasing insulin levels. According to the studies, this compound can lower glycolysis through p-AMPK activity impairment. The treatment group given 200 mg/kg of free anthocyanin compound of sweet potato (FAC-PSP) extract containing 40.74 ± 2.88 mg C3G/g for 4 weeks showed a substantial rise in the p-AMPK expression levels. In the liver, the insulin-responsive glucose transporter GLUT-2 is crucial for metabolism and glucose uptake. The manifestation of the GLUT-2 protein possesses the capability to amplify the re-uptake and usage of glucose in the liver [24]. Cyanidin 3-caffeoyl-p-hydroxybenzolsophoroside-5-glucoside and Peonidin 3-(6″-caffeoyl-6‴-feruloyl sophoroside)-5-glucoside have been found to restrain hepatic gluconeogenesis in HepG2 cells. However, the outcomes of an in vivo study investigating the effects of cyanidin revealed that oral administration significantly reduced fasting blood glucose levels from their initial high values at time zero (186–205 mg/dL, respectively [33].

According to earlier studies, it was found that blackcurrant extract, which consists of 45% anthocyanins and 82% total polyphenols, has potential to enhance plasma GLP-1 levels by approximately 30% and stimulate AMPK in the liver. Cyanidin 3-caffeoyl-p-hydroxybenzolsophoroside-5-glucoside and Peonidin 3-(6″-caffeoyl-6‴-feruloyl sophoroside)-5-glucoside have been reported to restrain hepatic gluconeogenesis in HepG2 cells. However, the outcomes of an in vivo study investigating the effects of cyanidin revealed that oral administration significantly reduced fasting blood glucose levels from their initial high values at time zero (186–205 mg/dL, respectively) [63,64]. The blocking of PEPCK and the G6Pase expression exhibited remarkable effectiveness in thwarting the escalation of blood glucose levels. This indicates that the process of gluconeogenesis will be suppressed, leading to a subsequent decrease in the production of glucose [65]. There is ongoing discussion regarding the mechanism by which GLP-1 affects hepatic gluconeogenesis and glycogen formation in the liver. Some argue that these effects are directly mediated by GLP-1R in hepatocytes, while others suggest that they may be indirectly mediated by the central nervous system (CNS) or insulin release. In vitro studies have shown that GLP-1 promotes glycogen synthesis and reduces gluconeogenesis by upregulating glycogen synthase, which is downstream of PI3K/PKB, PKC, and serine/threonine protein phosphatase 1. Additionally, GLP-1 decreased the expression of the gluconeogenetic enzyme phosphoenol pyruvate carboxykinase in rat hepatocytes [65].

The current research was primarily focused on the P13K/AKT pathway, which is considered to be one of the key insulin signaling pathways. [66]. Moreover, the PI3K/AKT/GSK-3ß signaling pathway activation not only improves insulin sensitivity and glucose metabolism, but also exerts a beneficial influence on dyslipidemia. A study conducted using high doses of sweet potato leaf polyphenols (SPLP) at 150 mg/kgBB demonstrated a more effective reduction in fasting blood glucose (FBG) over a period of 4 weeks compared to low-dose treatment. These findings suggested that the reduction in FBG by SPLP was both dose-dependent and time-dependent. The various components of SPLP, including 1-caffeoylquinic acid, 3,4,5-tricaffeoylquinic acid (3,4,5-triCQA), chlorogenic acid, caffeac acid derivative, quercetin, isoquercitrin, hyperoside, and rutin, play a crucial role in regulating hepatic glycogen synthesis in the liver. This regulation is mediated by insulin through the upregulation of PI3K/AKT and the downregulation of the GSK-3ß FOXO1 expression [11].

#### 4.2.5. Muscle

##### Enhancing the Absorption of Glucose, Secretion, and Insulin Sensitivity

The regulation of glucose metabolism and the maintenance of energy homeostasis heavily relies on the insulin-stimulated uptake of glucose in skeletal muscle. The PI3K/Akt pathway is a critical target for the treatment of type 2 diabetes mellitus (T2DM) due to its involvement in modulating the signaling pathways associated with muscle function [66]. In this study, it was hypothesized that the presence of arabinogalactan and epigallocatechin in WSPP was expected to result in an elevation in the expression degrees of p-IR, p-Akt, and M-GLUT-4. Remarkably, the administration of high doses of DM + 30% WSP-Tuber and DM + 5% WSP-Leave resulted in a significant reduction in fasting blood glucose levels, thereby improving the fasting glucose tolerance [23]. Epigallocatechin has recently been demonstrated to enhance the secretion of GLP-1. By overexpressing GLP-1, it will be able to enter the systemic circulation and bind to the GLP-1R receptor in different tissues, including skeletal muscle. Consequently, this binding event can lead to an elevation in the level of cAMP signaling through the Gs protein, thereby promoting AMPK phosphorylation. By promoting the translocation of GLUT-4 from its intracellular depot to the sarcolemma, this process effectively stimulates glycogen synthesis and enables the uptake of glucose in skeletal muscle [45,67].

Insulin induced a notable rise in AKT phosphorylation in another fraction within the ≤10 kDA range from WSSp [32]. The expression of GLUT-4, the NRF1 gene, and MERF2a was observed to increase after subjecting C2C12 skeletal muscle cells to tissue culture testing. In this experiment, the cells were subjected to different concentrations of OSPT (orange sweet potato tubers) and OSPL (orange sweet potato leaves) for a period of 3 h. The doses used were 500 µg/mL of OSPT and 100 µg/mL of OSPL. The expression of GLUT-4, a key factor in glucose absorption, was regulated by the transcription factors MEF2a and NRF1. The metabolism of glucose absorption was significantly influenced by these transcription factors. The close relationship between the expression and activity of the GLUT-4 gene and the NRF1 and MEF2a genes, as well as their association with insulin sensitivity and glucose homeostasis in skeletal muscle, has been discovered. ACC2 and CPT1 play crucial roles as regulators of mitochondrial fatty acid oxidation, and therefore, any interventions that affect their expression can impact intracellular lipid levels and have therapeutic implications in managing insulin resistance. The enhanced expression of these genes in the treated cells indicated that a compound derived from aqueous methanol extracts of orange-fleshed IBL, containing caffeaic acid, catechin, hysperoside, kaemferol, rutin, isovanillic acid, quercetin, protocatechui acid, and vanillic acid, holds the potential for enhancing insulin sensitivity [22].

#### 4.2.6. Adiposa

##### Increasing Glucose Uptake and Insulin Secretion

In vitro experiments have confirmed the effectiveness of a pure compound comprising quercetin 3-O-β-D sophoroside, quercetin, benzyl β-d-glucoside, 4-hydroxy-3-methoxybenzaldehyde, and methyl decanoate. This compound significantly increased the expression of PI3K, AKT, and GLUT-4 phosphorylation in 3T3-L1 adipocytes when tested at a dose of 0.01 mg/mL through a Western blot analysis. The activation of this gene increased GLUT-4 translocation so that glucose uptake also increased [13,68,69]. Caiapo containing aglycoprotein 4 g/day orally has been clinically tested on 30 patients and has been effective in reducing HbA1c progressively in diabetes patients for 1–2 months when compared to the placebo group [30]. Other studies also reported that caffeic acid could increase insulin secretion and sensitivity by increasing the adiponectin expression [31]. Adiponectin, a hormone that enhances insulin sensitivity and possesses anti-apoptotic and anti-inflammatory properties, is predominantly synthesized in adipose tissue. In individuals with obesity and type 2 diabetes, there has been a notable reduction in adiponectin levels. The administration of adiponectin has been shown to augment glucose uptake stimulated by insulin through the activation of AMPK in primary rat adipocytes. Moreover, adiponectin directly interacts with insulin receptor substrate-1 (IRS-1) and plays a pivotal role in facilitating insulin-mediated glucose uptake in adipocytes [70,71].

## 5. Conclusions and Future Perspective

IBL was reported to have variety and cultivars that contain chemicals with anti-diabetic properties. Such chemical compounds include flavonols, flavanol, flavones, antochyanin, phenolic acid, and triterpenoid groups. IBL can be considered as a multi-chemical and multi-pharmacological site since it functions in multiple organs through various ways. GLP-1 therapy for DM will prove to be quite advantageous in the future due to its efficacious nature. Flavonols, flavones, and flavone groups capable of forming complexes with AlCl_3_ can also form complexes with Ca^2+^ through a chelation mechanism. Therefore, it is crucial to scientifically establish the correlation between chelated AlCl_3_ + flavonoids, which have the potential to stimulate GLP-1 production. Additionally, it is important to determine whether the active AlCl_3_–flavonoid compound can inhibit the DPP-IV enzyme, as GLP-1 has a short half-life due to the degradation by this enzyme. The group compounds play a crucial role in increasing GLP-1 activity and exerting its anti-diabetic effects, which would subsequently strengthen insulin production and the uptake of glucose into cells as glycogen from the systemic circulation.

Consequently, this research presents significant challenges that necessitate rigorous scientific validation. Once this hypothesis is confirmed, it will pave the way for the development of new drugs with well-defined pharmacological mechanisms. The results of the ALCl_3_ + flavonoid complex require an isolation and structure elucidation process. It is hoped that the structure of the compound in IBL, a new drug candidate for treating DM, can be determined. Therefore, future research should involve in silico, in vitro, and in vivo testing of each isolated compound to strengthen the understanding of the pharmacological mechanisms involved. Subsequently, a chronic toxicity test should be conducted to ensure the safety and efficacy of the new drug candidate. The findings from this research are expected to be valuable for the pharmaceutical industry, which is interested in further developing it into drug formulations. Health professionals will gradually recognize that herbal medicines share similar pharmacological mechanisms with synthetic medicines, leading to an inclination towards prescribing herbal medicines for diabetes mellitus patients. Therefore, policymakers should consider legalizing the prescription of natural medicines among health professionals in order to provide patients with a sense of comfort when receiving DM medications made from herbal ingredients in the future.

## Figures and Tables

**Figure 1 metabolites-14-00029-f001:**
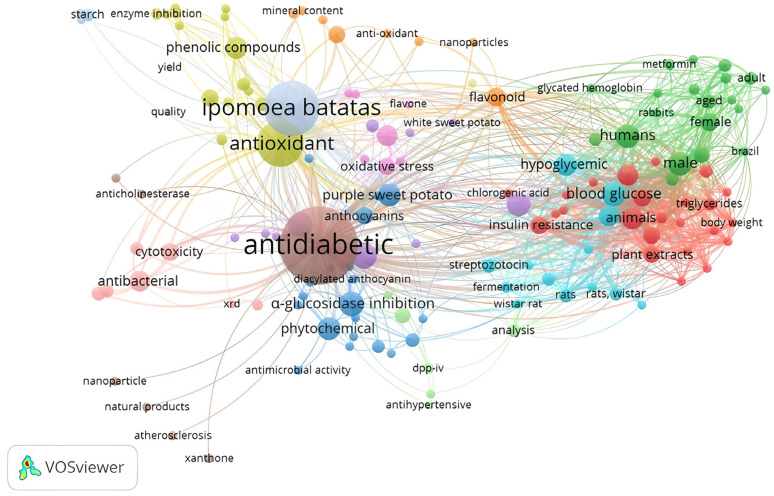
Keyword co-occurrence network.

**Figure 2 metabolites-14-00029-f002:**
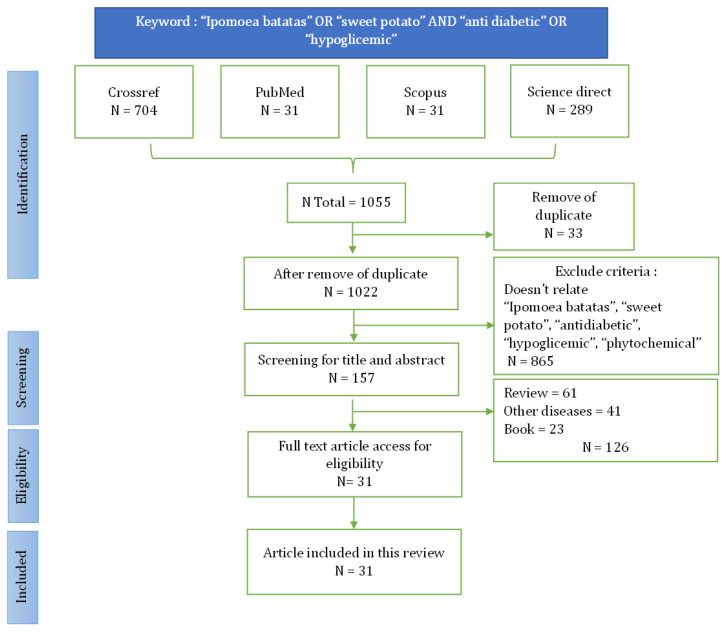
Flow chart identification and screening for the literature search.

**Figure 3 metabolites-14-00029-f003:**
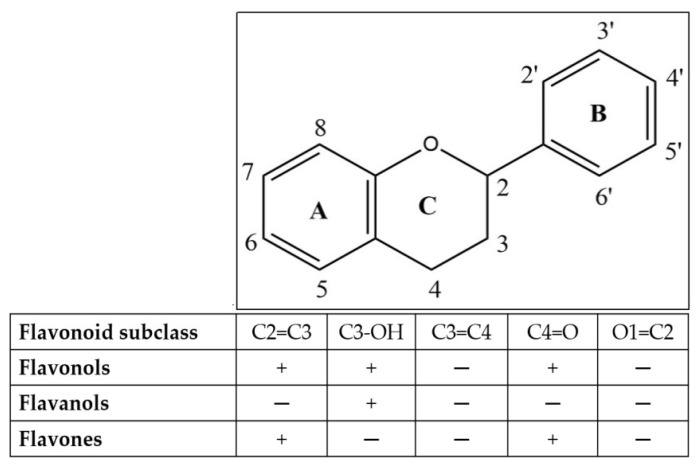
Flavonoid skeleton.

**Table 1 metabolites-14-00029-t001:** Type/cultivar of Ipomoea batatas, the predictive bioactive compounds, sites of action activity, and pharmacology mechanisms.

No	Type/Cultivar	Part of Plant	Identified Compound	Predictive Bioactive Compound	Analytical Method	Site of Action	Mechanism Pharmacology	Reference
1	IBL from cultivar Simon (Beijing, China)	Leaves	1-Caffeoylquinic acid; Neochlorogenic acid; Esculin; Protocatechualdehyde; Chlorogenic acid; Cryptochlorogenic acid; Caffeic acid hydroxycoumarin; Isochlorogenic acid A, B, and C, 3,4,5-Tricaffeoylquinic acid; Rutin, Hyperoside; Isoquercitrin; Astragalin; Quercetin; KAE; Diosmetin; Jaceosidin; Chrysin; and Pectolinarigenin	-	UHPLC-hybrid quadrupole-orbitrap/MS	Pancreas	Inhibiting beta cell apoptosis and recovering the islet structure	[11]
				-		Liver	Enhancing the AKT/PI3K/GSK-3ß signaling pathway leads to a decline in dyslipidemia, an enhancement in insulin sensitivity, and an enhancement in glucose metabolism	
				-		Muscle	The improvement of insulin sensitivity and the facilitation of glucose transport through upregulating the AKT/PI3K/GLUT-4 signaling pathway	
		Leaves	-	3,4,5-Tricaffeoylquinic acid; Cryptochlorogenic acid; Chlorogenic acid; Isochlorogenic acid A, B and C; Neochlorogenic acid; Esculin; Protocatechualdehyde; Caffeic acid; 7-hydroxycoumarin; Ethyl Caffeate; Rutin; Hyperoside; Isoquercitrin; Astragalin; Quercetin; Kampferol; Diosmetin; Jaceosidin; Chrysin; Pectolinarigenin; Hysperidin; Luteolin; and Catechin	UHPLC-hybrid quadrupole-orbitrap/MS	Gastrointestinal	α-amylase inhibition α-glucosidase inhibition	[12]
						Pancreas	Protection of cell beta by antioxidant capacity using ABTS, DPPH, FRAP	
2	Purple IBL from in Luzhu District, Taoyuan City, Taiwan	Leaves		Methyl decanoate: Quercetin 3-O-β-D sophoroside; 4-Hydroxy-3-methoxy benzaldehyde; Quercetin; and Benzyl β-d-glucoside	GC-MS	Adipose	GLUT-4 activation PI3K/AKT pathway regulation in 3T3-L1 adipocytes	[13]
3	IBL from the local market, India	Leaves		Acidic glycoprotein	-	Gastrointestinal	α-Glucosidase inhibition	[14]
4	IBL from Aan Village, Klungkung Regency, Bali Province, Indonesia	Leaves	-	Peonidin-caffeoyl-p-hydroxybenzoylsophorside-5-glucoside; Cyanidin 3-O-rutinoside (C3OR); Peonidin dirhamnoside; Cyanidin-3-glucoside isomer (C3G); Pelargonidin glucoside or cyanidin 3-O-rutinoside; and Peonidin dirhamnosaloyl-glucoside isomer	ESI-MS	Pancreas	Protection of pancreatic beta cell islet through the inhibition of intracellular nitric oxide (NO) and the reactive oxygen species (ROS) scavenging mechanism	[15]
5	Fresh orange-fleshed SPL (Jishu No. 16) collected from a farm in Yichun	Leaves	Trans-N-(p-coumaroyl) tyramine; 7,3′-Dimethylquercetin; 7-Hydroxy-5-methoxycoumarin; Caffeic acid ethyl ester; Trans-N-feruloyltyramine; Cis-N-feruloyltyramine; 3,4,5-Tricaffeoylquinic acid; 3,4-Dicaffeoylquinic acid; 4,5-Dicaffeoylquinic acid; 4,5-Feruloylcourmaoylquinic acid; Caffeic acid; Quercetin-3-O-a-D-glucopyranoside; and Indole-3-carboxaldehyde	3,4,5-Tricaffeoylquinic acid; 4,5-Dicaffeoylquinic acid; 3,4-Dicaffeoylquinic acid; Caffeic acid Quercetin-3-O-α-D-glucopyranoside; and 7,3′-dimethylquercetin	HPLC, 1D NMR, 2D NMR, ESI-MS	Pancreas	Protection of beta cell pancreas by decreasing ROS through radical scavenging DPPH	[16]
				Trans-N-(p-coumaroyl) tyramine; Trans-N-feruloyltyramine; 7-Hydroxy-5-methoxycoumarin; Cis-N-feruloyltyramine; Caffeic acid ethyl ester; 3,4,5-Tricaffeoylquinic acid; and Indole-3-carboxaldehyde		Gastrointestinal	α-Glucosidase inhibition	
6	IBL leaves from (Hebei province) in Autumn	Leaves	Flavone		-	Pancreas	The beta cell acts as a safeguard by efficiently eliminating excessive free radicals, consequently decreasing the occurrence of lipid peroxidation	[17]
7	IBL from Slatina (central Croatia)	Leaves	Flavonoid; Phenol		-	Pancreas	Protecting beta cell scavenging	[18]
8	IBL from Anguillara Veneta (Northern Italy)	Leaves	Catechin; Naringin; Epicatechin; Chlorogenic acid; p-OH benzoic acid; Vanillic acid; t-Ferulic acid; and o-Coumaric acid	Chlorogenic acid and Epicatechin	HPLC-PDA	Gastrointestinal	α-glucosidase inhibition Tyrosinase inhibition α-amylase inhibition	[19]
						Pancreas	Anti-apoptosis beta cells through suppressing the increasethe activation of caspase-3 and caspase-8	
9	Fresh leaves of IBL (family of clones B 00593) from Bandungan, Central Java Indonesia.	Leaves		Anthocyanins; Catechins; Quercetin; Proanthocyanidins; and Caffeic acid	-	Pancreas	The preservation of beta cells involves the suppression of reactive oxygen species (ROS) production, the removal of ROS through scavenging processes, and the augmentation of antioxidant defense mechanisms. In order to enhance insulin secretion and improve insulin sensitivity, it is crucial to inhibit the activation of NF-κB, thereby suppressing the production of TNF-α and inhibiting iNOS (inducible nitric oxide synthase).	[20]
10	‘Suioh,’ a IBL cultivar from Kumamoto prefecture, Japan	Leaves	Chlorogenic acid; 3,4,5-Tricaffeoylquinic acid; 3,4-Dicaffeoylquinic acid; 4,5-Dicaffeoylquinic acid; and 3,5-Dicaffeoylquinic acid	3,4,5-Tricaffeoylquinic acid	-	Gastrointestinal	α-glucosidase inhibition GLP-1 receptor activation, EGFRs are transactivated, thereby activating PI3K. This PI3K activation is essential for the cAMP/PKA-dependent pathway, which ultimately leads to the exocytosis of insulin. The exocytosis of insulin activates KATP channels, causing depolarization and an increase in Ca2+ levels, resulting in the secretion of insulin.	[9]
11	Purple IBL	-	Peonidin-3-glucoside (P3G); Cyanidin-3-rutinoside C3R); Cyanidin-3-glucoside (C3G); and Cyanidin-3,5-glucoside (C35G)	P3G; C3R; C3G; and C35G	-	Gastrointestinal	Inhibition porcine pancreatic α-amylase	[21]
12	‘Bophelo’ orange-fleshed IBL cultivar	Tubers and leaves	Isovanillic acid; Protocatechuic acid; Quercetin; Caffeic acid; Catechin; Hyperoside; Kaempferol; Rutin; and Vanillic acid	Isovanillic acid; Kaempferol; Protocatechuic acid; Caffeic acid; Catechin; Hyperoside; Rutin; Quercetin; and Vanillic acid	HPLC-MS	Pancreas	Protecting beta cell melalui peningkatan antioxidant enzyme (catalase, CAT, glutathione peroxidase) dan uji antioxidant capacity using FRAP and TEAC	[22]
						Muscle	Activation of GLUT-4 and improvement in glucose uptake Expression of the genes NRF1, MEF2A, CPT1, and ACC2, and ultimately glucose uptake metabolism and the management of insulin resistance	
13	White potato Tainung No.10	Tubers and leaves		Arabinogalactan; and Epigallocatechin gallate	-	Muscle	Activation of PI3K/Akt/GLUT-4 to increase insulin sensitivity and glucose uptake	[23]
14	Purple IBL (Cultivar Eshu No.12) from the Institute of Food Crops, Hubei Academy of Agricultural Sciences	Tubers	Peonidin-3-sophoroside-5-glucoside (P3S5G); Cyanidin-3-sophoroside-5-glucoside (C3S5G); Anthocyanins (containing one or two p-hydroxybenzoic, caffeic and/or ferulic acid); and 17 proteins (consisted of group: Acetylesterase, Proteinase inhibitor, Sporamin A, Superoxide dismutase [Cu-Zn], Beta-amylase, Sporamin B, preprosporamin, Polyphenol oxidase I chloroplastic, Purple acid phosphatase, and NBS-LRR protein and pectin)	P3S5G; C3S5G; Anthocyanins (containing one or two p-hydroxybenzoic, caffeic and/or ferulic acid); and 17 proteins (consisted of the group: Acetylesterase, Proteinase inhibitor, Sporamin A, Superoxide dismutase [Cu-Zn], Beta-amylase, Sporamin B, preprosporamin, Polyphenol oxidase I chloroplastic, Purple acid phosphatase, and NBS-LRR protein and pectin)	HPLC-DAD/ESI-MS	Pancreas	Protecting cell beta through reducing ROS and improving antioxidant enzyme activities	[24]
						Liver	Activation of AMPK/GLUT-2/GK and insulin receptor alfa (INSR) to increase the level of insulin and glucose transporter The synthesis of glucose is diminished through the downregulation of gluconeogenic genes, specifically glucose-6-phosphatase (G6Pase) and phospoenolpyruvate carboxykinase (PEPCK)	
15	Purple IBL powder (cultivar Eshu No. 8)	Tubers	Diacylated anthocyanins	Peonidin-3-caffeoylferuloyl sophoroside-5-glucoside	-	Liver	Enhancing the secretion and sensitivity of the insulin elucidates mechanism: (i) inhibitor of liver XO activity; (ii) activation of the expression of SGLT2, GLUT-5, and GLUT-2; (iii) the suppression of the NF-κB pathway leads to a decrease in the expression of IL-1ß and iNOS	[25]
16	*IBL (Linn.*) *Lam* from Western Research Farm, National Root Crop Research Institute, Umudike, Abia state	Tubers	Flavonoid; Terpenoid; Tannin; Phenol		-	Pancreas	Induction of beta cell regeneration or repairing and increasing the size and number of cells in the islet of Langerhans	[26]
						Gastrointestinal	α-glucosidase inhibition	
						Adipose	Activation of PI3K (Phosphoinositol-3-kinase), P38 MAPK (Mitogen-activated protein kinase), and GLUT-4 translocation. They have been seen to increase glucose uptake. Insulin secretagogues, directly activating the K+ ATP channel through influx of Na+ and an outflow of K+	
17	Purple IBL cv. Ayamurasaki from the Kyushu National Agricultural Experiment Station in Miyazaki prefecture (Japan)	Tubers		Peonidin 3-O-[2-O-(6-O-E-feruloyl-β-D-glucopyranosyl)-6-O-E-caffeoyl-β-D-glucopyranoside]-5-O-β-D-glucopyranoside	-	Gastrointestinal	α-Glucosidase inhibition	[27]
18	Korean red skin IBL (Ib 1)and Korean pumpkin IBL (Ib 2) from the market in Goyang, Republic of Korea	Peel-off tuber		α-carotene; ß-carotene; zeaxanthin; and lutein	-	Gastrointestinal	α-Glucosidase inhibition	[28]
19	White IBL (Caiapo)	Tubers		Acidic glycoprotein	-	Gastrointestinal	α-Glucosidase inhibition	[29]
						Adipose	The translocation of GLUT-4, along with the promotion of lipolysis and the subsequent release of free fatty acids from adipose tissue, contributes to an increased glucose uptake in isolated adipocytes. This mechanism ultimately results in a reduction in HbA1c levels.	[30]
20	White-skinned sweet potato (WSSP) purchased from Kagawa, Japan, Prefectural Cooperative	Tubers		Caffeic acid	-	Adipose	Improvement in the secretion and sensitivity of insulin through significant increases in the adiponectin expression	[31]
			WSPP fraction consists of >50 kDa, 10–50 kDa, and ≤10 kDa	≤10 kDa fraction	-	Muscle	Improving insulin sensitivity and glucose uptake. Considerably increases AKT phosphorylation//GLUT-4	[32]
				>50 kDa fraction		Liver	The inhibition of gluconeogenesis is achieved by suppressing the process itself and simultaneously promoting glycogen synthesis. This dual action leads to an increased uptake of glucose.	
21	Korean purple IBL (Shinzami, Saeungbone9, Gyeyae2469, Gyebone108, Saeungyae33, and Gyeyae2258)	Tubers	3-Caffeoyl-phydroxybenzoylsophoroside-5-glucoside; Peonidin 3-caffeoyl sophoroside-5-glucoside; Peonidin 3-(6″-caffeoyl-6‴-feruloyl sophoroside)-5-glucoside; and Peonidin 3-caffeoyl-phydroxybenzoylsophoroside-5-glucoside	Cyanidin 3-caffeoyl-p-hydroxybenzolsophoroside-5-glucoside and Peonidin 3-(6″-caffeoyl-6‴-feruloyl sophoroside)-5-glucoside	LC-DAD-ESI/MS	Liver	Inhibition of hepatic gluconeogenesis in HepG2 cells can lead to an enhancement in insulin sensitivity by reducing glucose secretion Protective beta cell by reducing ROS through radical scavenging	[33]
22	Color-fleshed potatoes (Sinjami and Sinhwangmi)	Tubers	Lutein; Peonidin 3-(6″-caffeoyl-6″-feruloyl sophoroside)5-glucoside; Zeaxanthin; Cryptoxanthi; 13Z-ß-carotene; Peonidin 3-sophoroside-5-glucoside; Peonidin 3-p-hydroxybenzoyl sophoroside-5-glucoside; Cyanidin 3-p-hydroxybenzoyl sophoroside-5-glucoside; Cyanidin3-(6″-feruloyl sophoroside)-5-glucoside; Peonidin 3-(6″-feruloyl sophoroside)-5-glucoside; Cyanidin 3-(6″,6″-dicaffeoyl sophoroside)-5-glucoside; Cyanidin 3-caffeoyl-p-hydroxybenzoyl sophoroside-5-glucoside; Cyanidin3-(6″-caffeoyl6″-feruloyl sophoroside)-5-glucoside; Peonidin 3-caffeoyl sophoroside-5-glucoside; Peonidin 3-(6″,6″-dicaffeoyl sophoroside)-5glucoside; Peonidin 3-caffeoyl-p-hydroxybenzoyl sophoroside-5-glucoside; all-trans-ß-carotene; and 9Z-ß-carotene	Peonidin 3-caffeoyl-p-hydroxybenzoyl sophoroside-5-glucoside	UPLC-MS/MS (Q-TOF-ESI)	Adipose	Stimulating adipogenesis through the inhibition of fat accumulation in adipocytes via the PPARγ expression. Activation of the PPARγ receptor will maintain glucose homeostasis	[34]
23	IBL from Kagawa Prefecture, Japan	Tubers		Chlorogenic acid; and Caffeic acid and its derivatives	-	Pancreas	Protection of beta cell from oxidative stress-related gene expression and peroxidation of the plasma membrane Increase in the secretion insulin by the inhibited activation of the nuclear transcription factor and P38 MAP kinase pathway to decrease TNF-α production	[35]
24	White-skinned sweet potato (WSSP)	Tubers		Arabinogalactan protein	-	Liver	Improving insulin sensitivity by the inhibition inflammatory cytokines such as IL-6 and TNF-α	[36]
25	Purple IBL Antin-3 cultivar from the BALITKABI Malang	Tubers	Anthocyanin group		-	Pancreas	Regeneration and protecting beta cells through reducing oxidative stress using the radical scavenging DPPH method	[37]
26	White-skinned sweet potatoes (WSSP) from the local market Faisalabad (Pakistan)	Tubers		Carotenoid	-	Pancreas	Protecting beta cells by decreasing oxidative stress, and induced elevated cytosolic free Ca^2+^ concentrations in beta cells further contribute to supraphysiological insulin release	[38]
				Glicoprotein; Flavonoid; and Carotenoid	-	Liver	Hepatoprotective mechanism due to a decrease in the glycation level prevents the formation of ROS—amplified activities of liver enzymes such as SGOT and SGPT.	[39]
27	Purple IBL from Padang, West Sumatra, Indonesia	Tubers		Peonidin; and Cyanidin	-	Gastrointestinal	α-amylase inhibition	[40]

## Data Availability

The data to support the finding of this study can be made available by the corresponding authors upon request. The data are not publicly available due to privacy or ethical restrictions.

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
