# Peer review of "The Potential of the Flavonoid Content of Ipomoea batatas L. as an Alternative Analog GLP-1 for Diabetes Type 2 Treatment—Systematic Review"

_metabolites, 2023, doi:10.3390/metabo14010029_

Round 1
Reviewer 1 Report
Comments and Suggestions for Authors
metabolites-2780083
"The Potential of Ipomoea batatas L. as an Alternative Analog GLP-1 for Diabetes Type 2 Treatment – A Systematic Review"
The authors have carried out an extensive data search on the subject of the review. In my opinion, the review is very well prepared and presents the results on a very important topic. This manuscript can be published in its present form. It will be useful for scientific readers.
Author Response
Thank you for the appreciation given to our article

Reviewer 2 Report
Comments and Suggestions for Authors
The authors report an overview on the potential application of Ipomoea batatas as an alternative analog 2 GLP-1 for the treatment of diabetes type 2.
The title needs to be improved because it is the metabolites of I.batatas that are responsible for this activity.
In the search strategy, it must be expressly indicated if any patents are included and the wording “6. Patents” on line 477 must be modified or eliminated accordingly.
As can be seen from figure 1, there are 31 articles included in the review and it is not clear why there are 38 references in the table. On the other hand, a certain number of reviews on similar topics have already been reported; e.g. from PUBMED search of review articles published from 2020 these results are obtained: a) for the keywords Ipomoea batatas + diabetes, 7 publications, b) for flavonois+DPP-IV: 4, and c) for flavonoids + diabetes: 175. Therefore, this review must be strictly focused on the topic described in the title (GLP-1), or it is redundant. In the mechanism pharmacology reported in table 1, GLP-1 is indicated only in the entry 10, and among the cited references it is included in the titles of 5, 18, 42-47.
On the presence of metabolites indicated in the table, the authors must add a short description of the analytical methods used to characterize the chemical profile, according to the indications in the original articles. Similarly, for the abundance of these components and biological evaluations carried out on purified metabolites.
Minor comments:
Line 234: which units for IC50 values?
Line 54, specify the full name for the acronym DM
In table 1, page 6, specify the reason for reporting the names of some metabolites in italics.
Author Response
We have rewritten the tittle “The Potential of Flavonoid Content of Ipomoea batatas L. as an Alternative Analog GLP-1 for Diabetes Type 2 Treatment – Systematic Review”
We have deleted the wording “6. Patents” on line 506-508
We selected the number 38 based on the information provided in table 1, specifically in column 1, starting from reference [8]. We ensured that a total of 31 articles were included, resulting in a cumulative count of 38 references.
In this systematic review we use the keywords Ipomoea batatas OR sweet potato AND anti-diabetic OR hypoglycemic. Adding the keyword “GLP-1” to our content would not yield any results on search engines. Consequently, we have made the decision to refrain from utilizing the GLP-1 keyword.
Our investigation has revealed one study specifically addressing GLP-1, which can be found in column 10. Nevertheless, references 5, 18, and 42-47 provide additional support by demonstrating that various compounds discussed in this systematic review also possess GLP-1 activity.

Reviewer 3 Report
Comments and Suggestions for Authors
The article comprehensively reviews the anti-diabetic properties of Ipomoea batatas L. (IBL), also known as sweet potato. The authors have conducted an extensive literature search following PRISMA guidelines, using Crossref, Pubmed, Scopus, and Science Direct databases. The search was focused on the anti-diabetic or hypoglycemic properties of IBL. The review reveals that IBL contains various compounds, including phenolic acid, flavonols, flavanols, flavones, and anthocyanins, which exhibit activity against diabetes. These compounds can form complexes with AlCl3 and Ca2+, leading to the retention of Ca2+ within intracellular L cells and the subsequent release of GLP-1.
Furthermore, the flavonols, flavones, and flavone groups in IBL have been found to interact strongly with DPP-IV, inhibiting the degradation of GLP-1. This mechanism effectively prolongs the half-life of GLP-1 in the systemic system, thereby contributing to the anti-diabetic activity of IBL. The article also mentions that IBL has a variety of cultivars containing chemicals with anti-diabetic properties, including flavonols, flavanol, flavones, anthocyanin, phenolic acid, and triterpenoid groups. IBL is considered a multi-chemical and multi-pharmacological site as it functions in multiple organs in various ways. The review concludes that GLP-1 therapy for diabetes mellitus (DM) will be advantageous in the future due to its efficacious nature. The compounds in IBL play a crucial role in increasing GLP-1 activity and exerting its anti-diabetic effects, which subsequently strengthen insulin production and glucose uptake into cells as glycogen from the systemic circulation.
Overall, the article provides a detailed and insightful examination of the anti-diabetic properties of IBL, highlighting its potential as a complementary therapy or herbal medicine in treating diabetes. The findings suggest that further research into the specific mechanisms of these compounds could lead to new therapeutic approaches for diabetes management.
Some potential questions/suggestions that could help improve the review article:
- Could the article provide more detailed information on the specific flavonoid compounds in IBL that contribute to its anti-diabetic properties?
- Could the article elaborate more on how these compounds form complexes with AlCl3 and Ca2+?
- How does the retention of Ca2+ within intracellular L cells lead to the release of GLP-1? Could this mechanism be explained in more detail?
- The article mentions that IBL functions in multiple organs in various ways. Could it provide specific examples or case studies to illustrate this?
- Could the article provide more evidence or research findings to support the claim that GLP-1 therapy for DM will be advantageous in the future?
- How do the compounds in IBL increase GLP-1 activity, and how does this lead to strengthened insulin production and glucose uptake?
- Could the article discuss potential side effects or risks associated with using IBL as a complementary or herbal medicine in treating diabetes?
- Are there any limitations or challenges in using IBL for diabetes management that the article could address?
- Could the article discuss the implications of these findings for patients, healthcare providers, and policymakers?
- Could the article suggest areas for future research based on the findings of this review?
- It is suggested to mention the importance of synthetic efforts towards development of antidiabetic medicine. In this regard, it is recommended to emphasis the importance of iminosugars and sugar derivatives as an antidiabetic agent and suggested to cite following relevant articles in the introduction section; i) https://doi.org/10.1002/anie.202217809 ii) Compain, P.; Martin, O. R. Iminosugars: From synthesis to therapeutic applications; Wiley-VCH:New York, 2007; pp 187−298 and iii) https://doi.org/10.24820/ark.5550190.p011.809.
Please add the relevant suggestions above with respective references to improve the article readership.
Author Response
Epicatechin, hyperoside, quercetin, and cyanidin-3-O-glycoside are flavonoids that have been examined for their anti-diabetic properties. These compounds have shown potential in regulating blood sugar levels through the GLP-1 mechanism. Their effectiveness has been evaluated both in vitro using the GluTag cell line and in silico.
the other respons you can find on attachmen
